

# Comparative analysis of automatic gender detection from names: evaluating the stability and performance of ChatGPT *versus* Namsor, and Gender-API

Adrián Domínguez-Díaz[1], Manuel Goyanes[2], Luis de-Marcos[1] and Víctor Pablo Prado-Sánchez[1]

[1] Ciencias de la Computación, Universidad de Alcalá, Alcalá de Henares, Spain
[2] Comunicación, Universidad Carlos III de Madrid, Getafe, Spain

## ABSTRACT

The gender classification from names is crucial for uncovering a myriad of gender-related research questions. Traditionally, this has been automatically computed by gender detection tools (GDTs), which now face new industry players in the form of conversational bots like ChatGPT. This paper statistically tests the stability and performance of ChatGPT 3.5 Turbo and ChatGPT 4o for gender detection. It also compares two of the most used GDTs (Namsor and Gender-API) with ChatGPT using a dataset of 5,779 records compiled from previous studies for the most challenging variant, which is the gender inference from full name without providing any additional information. Results statistically show that ChatGPT is very stable presenting low standard deviation and tight confidence intervals for the same input, while it presents small differences in performance when prompt changes. ChatGPT slightly outperforms the other tools with an overall accuracy over 96%, although the difference is around 3% with both GDTs. When the probability returned by GDTs is factored in, differences get narrower and comparable in terms of inter-coder reliability and error coded. ChatGPT stands out in the reduced number of non-classifications (0% in most tests), which in combination with the other metrics analyzed, results in a solid alternative for gender inference. This paper contributes to current literature on gender detection classification from names by testing the stability and performance of the most used state-of-the-art AI tool, suggesting that the generative language model of ChatGPT provides a robust alternative to traditional gender application programming interfaces (APIs), yet GDTs (especially Namsor) should be considered for research-oriented purposes.

Corresponding author
Luis de-Marcos,
luis.demarcos@uah.es

## INTRODUCTION

The substantial proliferation of open datasets from public and private scientific endeavors offers new opportunities to empirically explore a myriad of gender-related research questions (*Bérubé et al., 2020*; *VanHelene et al., 2024*). However, to fully understand the potential effect of gender on different human perceptions, behavior or attitudes, research needs to first provide and then test the performance of clear-cut gender detection tools to

classify gender of observations from names (*Bérubé et al., 2020*; *Sebo, 2021a*). Different examples of such empirical efforts may include, for instance, the effect of gender in scientific production, scientific representation, or citations, among many other socially relevant aspects (*Jung et al., 2018*; *Beaudry & Larivière, 2016*; *Holman, Stuart-Fox & Hauser, 2018*; *Astegiano, Sebastián-González & Castanho, 2019*; *Cimpian, Kim & McDermott, 2020*; *Goyanes et al., 2022*).

Research has tested and compared the performance of different gender classification tools and machine learning techniques (*Karimi et al., 2016*; *Bérubé et al., 2020*; *Ani et al., 2021*; *Sebo, 2021a*, *2022*; *Sánchez, Moreno & López, 2022*; *VanHelene et al., 2024*), providing relevant empirical evidence of the best gender classification tools, and suggesting the best course of action to address their potential limitations (*Ani et al., 2021*; *Sebo, 2021a*; *Sebo, 2022*; *Sánchez, Moreno & López, 2022*). Despite this vast literature, three main research gaps remain. Firstly, most studies computationally test the classification power of different gender detection tools directly, thus empirically trusting the reliability of the gender detection tools (GDTs) in inferring the correct gender. However, in absence of a ground truth against to which compare the gender classification output, the results of such inferences may be misleading. Although a few studies report and compare against the truth, they mostly focus on comparing general GDTs' performance (*Santamaría & Mihaljević, 2018*) or are circumscribed to specific geographies (*Sebo, 2021a*, *2021b*, *2022*). Accordingly, this study further covers this gap in the literature by testing the performance of two GDTs and one artificial intelligence (AI) tool against a dataset of empirically labeled gender-coded observations extracted from previous research, thus offering solid suggestions about their performance.

Secondly, while existing research has empirically tested and compared the gender classification performance of various GDTs specifically designed for that purpose (such as Namsor, Gender-API, GenderWiki Sort, *etc.*), the potential power of generative language models, such as ChatGPT, in predicting gender from names remains largely unexplored. To date, only two initial exploratory works (*Alexopoulos et al., 2023*; *Sebo, 2024*) have focused on the use of ChatGPT for gender detection. *Alexopoulos et al. (2023)* conducted a comparative study on the performance of ChatGPT and two GDTs using a dataset of over 134,000 Olympic athletes. Their analysis focused on variations in name input and the geographical origins of names. Their findings revealed that ChatGPT's performance was comparable to that of Namsor, even outperforming it in certain scenarios. However, in our opinion, they reported machine-learning metrics, which are not common practice in gender inference studies. In a subsequent study, *Sebo (2024)* evaluated the performance of ChatGPT 3.5 and ChatGPT 4.0 on a database containing 6,121 individuals. The study concluded that both versions of ChatGPT demonstrated high accuracy and a low rate of misclassifications. Furthermore, there was almost perfect inter-rater agreement between the two versions of the AI model. Sebo compared the results of ChatGPT with other studies that used similar datasets. Although existing studies point to the potential of ChatGPT for gender detection, the effect of the non-deterministic nature of large language models on the stability of gender detection remains unexplored. Further, the specific prompt may also impact on the results. Therefore, the contributions of this paper to the field are threefold:

the systematic testing of the stability of two versions of ChatGPT, the evaluation of performance under different prompting conditions, and the comparison of results with existing GDTs (Namsor and Gender-API).

Finally, thus far, prior literature has mainly provided results on classifications, missclassifications, nonclassification, errors, and gender bias of gender inferences (*Bérubé et al., 2020*; *Sebo, 2022*; *VanHelene et al., 2024*; *Alexopoulos et al., 2023*). However, beyond these relevant statistics, which are indeed very common in computational sciences and gender detection, prior literature has typically neglected the computation of inter-coder reliability measures to address the reliability of the gender predictions. Accordingly, this study computes the Krippendorff alpha and the Cohen's kappa, two of the most used and trusted statistics to empirically test the coding performance of content analyses. They measure how much we can trust the consistency of the inference made by different human coders or machine tools.

## METHOD

### Data

Data for this study was taken from *Santamaría & Mihaljević (2018)* who compiled a dataset of 7,076 records from previous bibliometric studies on gender patterns (*Larivière et al., 2013*; *Filardo et al., 2016*; *Mihaljević-Brandt, Santamaría & Tullney, 2016*) and a study on name-to-gender inference (*Wais, 2016*). Data includes only names labeled by gender, and the dataset of origin. The original sources and their respective gender coding methodologies are as follows:

1) zbMATH: This source provided bibliographical records of mathematical publications. The gender of the authors was determined using internet queries to gather gender information from author profile information from zbMATH and other bibliographic databases, university websites, Wikipedia articles, and similar online sources.

2) genderizeR: This package provided records of articles from all fields of study, published from 1945 to 2014. The gender of the authors was determined based on results from internet queries using the authors' full names, affiliations, biographies, mentions in the press, and photos.

3) PubMed: This source provided records from the six journals with the highest JCR impact factor in 2012 in the category Medicine, general & internal for original research articles between 1994 and 2014. The gender of the authors was determined by first inspecting the forename and then searching institutional websites, social media accounts, photographs, and biographical paragraphs for more information.

4) WoS: This source provided records from the WoS database covering all publications from 2008 to 2012. The gender of the authors was manually assigned based on biographical information located using the specific country, institution, and, in some cases, an email address associated with each name.

The final data set consists of 7,076 names (3,811 male, 1,968 female, 1,297 unknown), split into three components: first, middle, and last name. In addition to the gender coding,

*Santamaría & Mihaljević (2018)* also analyzed the geographic representation based on names. They concluded that the data originates from 113 countries from all main geographies, although European names are overrepresented. They were able to assign a provenance to 97% of the names using the Namsor's onomastics service. According to Namsor's estimates, 4,228 names (61%) were of European origin, 2,304 (34%) were of Asian origin, and 334 (5%) were of African provenance. This comprehensive dataset provides a robust foundation for further research on gender representation in academic publications. We preprocessed the dataset to remove all non-labeled records ($N = 1,297$). The final dataset used in this study includes 5,779 names ($N_{male} = 3,811$, $N_{female} = 1,968$).

## Tools

This study compared three methods of gender inference: Gender-API, Namsor and ChatGPT. Gender-API is a gender inference tool that provides services to infer gender from name, country, and email address. It also provides a service to infer country of origin from name. Namsor software provides services to classify names by gender, country of origin and ethnicity. Both Gender-API and Namsor provide an easy-to-use web site to upload and process data as well as application programming interface (API) to process data computationally. Namsor and Gender-API are commonly used and recognized for their performance in gender detection in different studies. Namsor, for instance, has been found to have a 0% rate of unrecognized names, the lowest among the four gender detection tools evaluated, while also showed a low level of errors (*Sebo, 2021a*). Namsor's machine learning algorithms and large databases enable it to predict whether a name is more likely to be masculine or feminine. The accuracy of gender determination can be improved by adding a country of residence. On the other hand, Gender-API offers efficient solutions for validating names, verifying surnames, and determining male or female gender from names with surnames. It optimizes processes such as auto-completion, name and surname verification, and spelling and syntax validation in systems of all kinds. It uses algorithms and extensive databases to predict whether a name is more likely to be masculine or feminine. It provides a significant level of accuracy that is crucial in many digital contexts. Previous studies compared them to a number of other existing tools concluding that they offer the best performance (*Santamaría & Mihaljević, 2018*; *Sebo, 2021a*), even for Asian names (*Sebo, 2022*). In conclusion, both Namsor and Gender-API provide robust and efficient solutions for gender detection, making them excellent choices for any research work. They offer a high level of accuracy, support for multiple languages, and the ability to handle large databases, which are essential features for any gender detection tool. Their superior performance in gender detection makes them the preferred choice for researchers in this field.

To utilize Namsor for gender identification, we first upload our data file, which can be in Excel or CSV format. We then choose the option that corresponds to our data under the "Gender from names" feature. Namsor offers a variety of options for instances where only the full name is available, or the country is coded. Next, we adjust the settings based on our needs. This includes deciding if we want to retain the original columns in the dataset, specifying if the dataset includes a header to identify the first data row, and identifying the

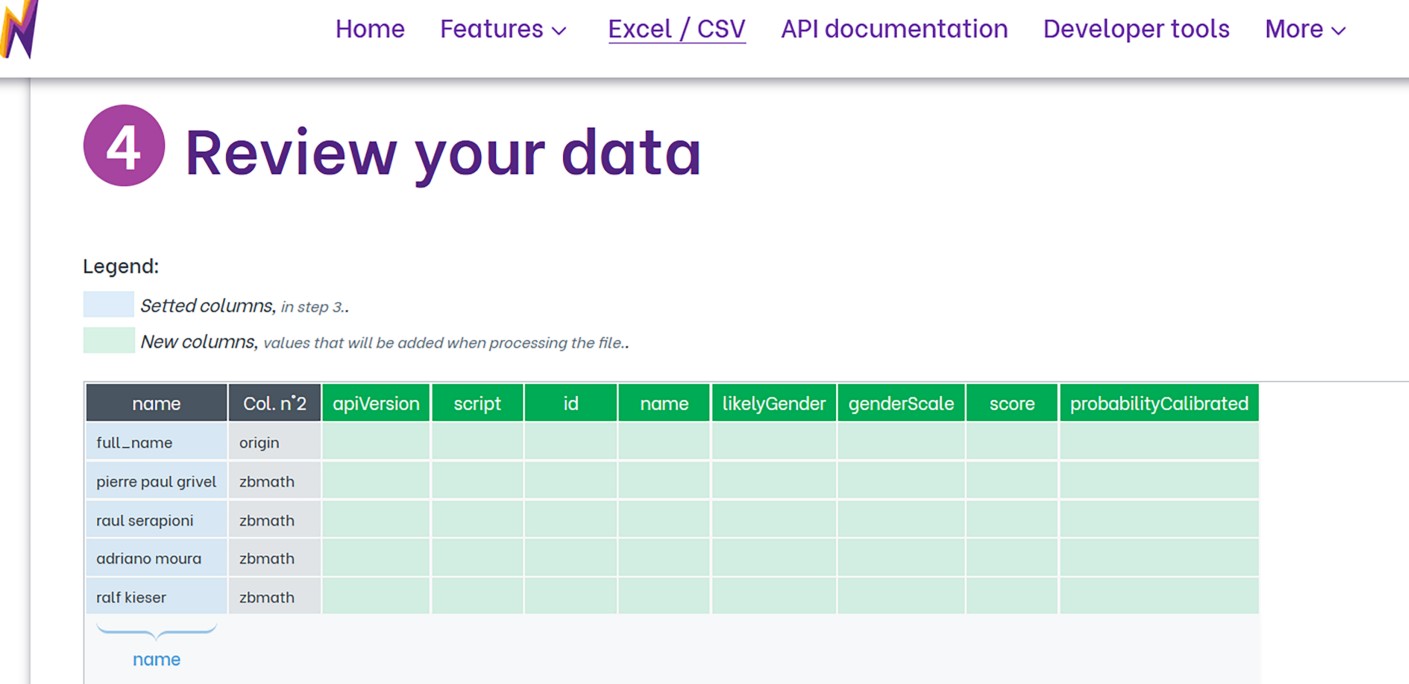

**Figure 1** Review screen from Namsor that shows the columns used to infer gender (in blue) and the new columns added with the prediction (in green). Columns not changed are shown in grey.

columns that represent the first name, last name, and the country code used for gender inference. Namsor then verifies the chosen columns. Subsequently, Namsor displays a data review screen that showcases the columns set up for prediction and the newly created output columns (as shown in Fig. 1). After a brief summary is shown, the file undergoes processing and is then ready for download. Namsor provides a user-friendly web interface for the entire process.

In a similar fashion, to deduce gender using Gender-API, we initiate by uploading the file, choosing either CSV Upload or Excel Upload. We then select the fields corresponding to the first or full name, and the country code if it's available. Upon clicking 'review your data', Gender-API presents the result of processing the initial ten records (as depicted in Fig. 2). We have the option to modify the previous settings or proceed to process the dataset by clicking on the designated button. Once the button is clicked, Gender-API begins processing the file which can subsequently be downloaded. Given that Gender-API is solely designed to infer gender, its interface is more straightforward compared to Namsor's.
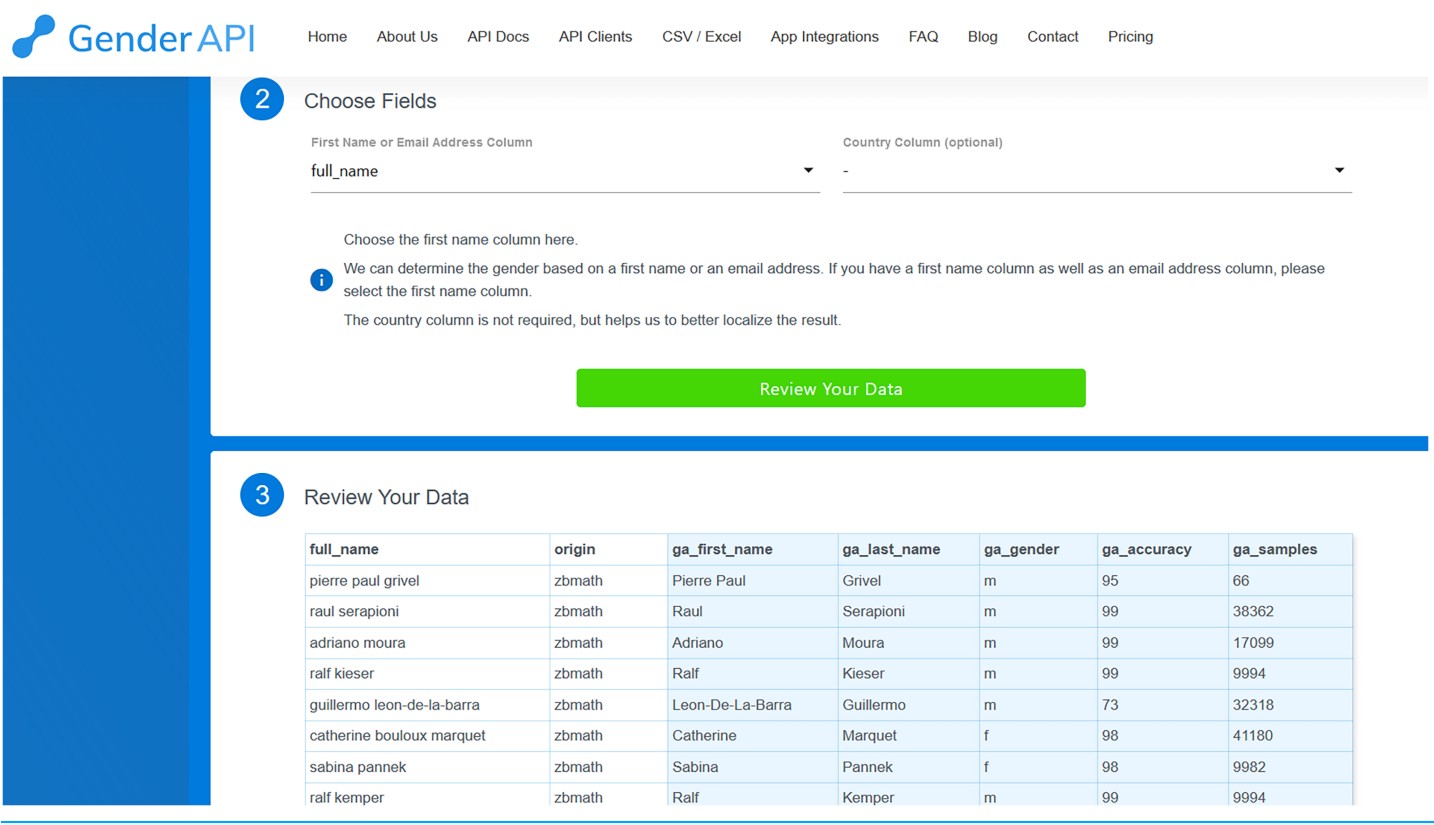

**Figure 2** Review screen from Gender-API that shows the original columns and the new columns added with prediction (in light grey).

ChatGPT is a general-purpose conversational bot, that can be accessed through a visual web interface for individual queries, or through a code-based API for bulk requests, such as dataset processing. We can use a direct prompt to ask for the gender of a given name (*e.g.*, "Jane Smith is a male or female name?"). ChatGPT web interface limits the size of the output to 4,096 tokens, so we programmed a script to process a dataset of an arbitrary size through the API and collect the results in a spreadsheet (Table 1).

Accuracy of GDTs improves if first and last names are separated. GDTs are also more accurate if additional information, like the country of residence, is provided. Both Gender-API and Namsor provide functions for processing split names and country of origin. In ChatGPT, we can also use a prompt to separate the name or include the country of origin in the query (*e.g.*, "Is Jane Smith from Australia a male or female name?"). This study analyses gender inference from full name, which is the most challenging scenario. In Namsor, we used the function 'Gender from unsplit full name' with the version 2.0.29 of the API. We used Gender-API function 'Gender from full name' with version 2.0. For ChatGPT, we used the script provided in Table 1 with two different versions: ChatGPT 4o and ChatGPT 3.5 Turbo. We chose the last version (gpt-4o-2024-05-13), at the time of writing this paper, and the last iteration of the previous version (gpt-3.5-turbo-0125),

**Table 1 Source code for gender inference using ChatGPT.**

```python
# Import the pandas library to work with DataFrames
import pandas as pd
# Import the OpenAI class from the openai library
from openai import OpenAI
# Authenticate with OpenA API
client = OpenAI(api_key = "YOUR API KEY HERE")
# Read the .xlsx file and store the data in a DataFrame
df = pd.read_excel(r"YOUR .XSLX SOURCE FILE PATH HERE")
# Iterate over the rows of the DataFrame and send the content
# of each row as input to the ChatGPT API
responses = []
for index, row in df.iterrows():
    prompt = f"{\
        str(row['FullName']).strip()}" ". \n\n \
        On a first and last name basis, \
        give me a letter answer only, \
        f if female; m if male. \n\n \
        Return the response in JSON format with key 'gender'."
    response = client.chat.completions.create(
        model="gpt-4o", # Selects the OpenAI API model to be used
        messages=[
            {"role": "system", "content": prompt}
        ],
        response_format={ "type": "json_object" }
    )
    response = response.choices[0].message.content.strip()
    response_json = json.loads(response)
    responses.append(response_json["gender"])
# Add answer column to original DataFrame
df["Responses ChatGPT"] = responses
# Save DataFrame with answers in .xlsx file
df.to_excel(r"THE PATH TO YOUR FINAL .XSLX FILE HERE", index=False)
```

**Note:**
Code was run with ChatGPT 4o and ChatGPT 3.5 Turbo and calls the OpenAI API v1.12.0.

which is much cheaper. Earlier versions can be even freely used if run on the appropriate hardware. Predicted gender was coded as 'f' for female or 'm' for male. Since gender inference cannot determine the predicted gender in all cases, non-classifications were coded as 'u'.

GDTs also return the probability of the prediction, so we also collected Namsor's calibrated probability and Gender-API's accuracy. Probability of prediction can be used to control and fine-tune the output of the genderization. Particularly, the gender of unisex or gender-neutral names is difficult to infer. Since GDTs infer gender based on the number of instances stored in their internal databases, the probability can be used as a measure of the certainty of the prediction, for example by establishing a threshold. Namsor documentation suggests that all predictions with a probability equal or lower than 0.55 can be consider unisex names. Gender-API returns a similar measure called accuracy which ranges between 50 and 99. For this study we coded both cases for both GDTs having, (1) Namsor without calibrated probability, Namsor when probability is calibrated to 0.55 ($Namsor_{p > 0.55}$), Gender-API without calibrated accuracy, and Gender-API when accuracy is higher than 55 ($Gender\text{-}API_{acc > 55}$). All instances in which the probability was less or equal than the threshold were coded as unknown.

At the time of writing, ChatGPT cannot provide a similar useful measure of probability. We tried different prompts but, in all cases, the best we could get was a two-rank probability ('medium' and 'high'). When asked for a numerical value, it always returned either 0.75, and only in a few cases, or 0.99. Authors did not consider this as an equivalent measure of probability of prediction, so it was not included in the study. Therefore, GDTs such as NamSor and GenderAPI, currently present a methodological advantage over ChatGPT. These tools provide a calibrated probability measure for their predictions, which is a valuable feature for researchers. This measure allows for a more nuanced understanding of the confidence level of the gender inference, which can be factored into experiments and analyses.

In terms of pricing, at the time of writing, Gender-API offers both monthly subscriptions and one-time payment packages. The pricing starts at approximately $7.4 for a 5,000-request monthly subscription. On the other hand, NamSor only offers subscription services, with prices starting at $19 for 10,000 monthly requests. However, it also provides 5,000 monthly requests for free. The cost of using ChatGPT *via* its API is calculated based on the runs described below, with the code and prompt presented in Table 1. For 10,000 requests, the cost is $0.41 for ChatGPT 3.5 and $4.07 for ChatGPT 4o. Please note that since ChatGPT's pricing is based on the number of tokens, the prompt and output format may result in variations in costs.

### Metrics

Gender inference is a two-class (male, female) classification problem in which the predicted class can also include non-classifications (male, female, unknown). The confusion matrix presents six measures: the number of males correctly classified as males ($m_m$), the number of females correctly classified as females ($f_f$), the number of males incorrectly classified as females ($m_f$), the number of females incorrectly classified as males ($f_m$), the number of males classified as unknown ($m_u$) and the number of females classified as unknown ($f_u$). For this study, we built the confusion matrix for each case (Namsor, $Namsor_{p > 0.55}$, GenderAPI, and $Gender\text{-}API_{acc > 55}$, ChatGPT 4o and ChatGPT3.5) and computed the following metrics: accuracy, overall error, error coded without NA, NA

**Table 2 Formulas for the metrics to assess gender-inference performance.**

| Metric | Formula |
|---|---|
| Accuracy | $\dfrac{m_m + f_f}{m_m + f_m + m_f + f_f + m_u + f_u}$ |
| Error coded | $\dfrac{f_m + m_f + m_u + f_u}{m_m + f_m + m_f + f_f + m_u + f_u}$ |
| Error coded without NA | $\dfrac{f_m + m_f}{m_m + f_m + m_f + f_f}$ |
| NA coded | $\dfrac{m_u + f_u}{m_m + f_m + m_f + f_f + m_u + f_u}$ |
| Gender bias | $\dfrac{m_f - f_m}{m_m + f_m + m_f + f_f}$ |
| Cohen's kappa | $\dfrac{D_0 - D_e}{1 - D_e}$ where $P_0 = \dfrac{m_m + f_f}{m_m + m_f + f_m + f_f}$ and $D_e = \dfrac{(m_m + m_f)(m_m + f_m) + (f_m + f_f)(m_f + f_f)}{(m_m + f_m + m_f + f_f)^2}$ |
| Krippendorff's alpha | $1 - \dfrac{D_0}{D_e}$ where $D_0 = m_f + f_m$ and $D_e$ as defined in previous formula |
| F1-score | $F1 - score_f = 2 \cdot \dfrac{Precision_f \cdot Recall_f}{Precision_f + Recall_f}$ where $Precision_f = f_f / f_f + m_f$ and $Recall_f = f_f / f_f + f_m$ |

**Note:**
$m_m$ is the number of males correctly classified as males, $f_f$ is the number of females correctly classified as females, $m_f$ is the number of males incorrectly classified as females, $f_m$ is the number of females incorrectly classified as males, $m_u$ is the number of males classified as unknow and $f_u$ is the number of females classified as unknow. Initial four metrics are taken from *Wais (2016)* and *Santamaría & Mihaljević (2018)*. F1-score was computed for both classes ('m' and 'f') and averaged (macro F1-score). The table presents only the formula for the 'f' class. It is analogous for the 'm' class.

coded, gender bias, Cohen's kappa, Krippendorff's alpha, and F1-score. Formulas are provided in Table 2.

The initial five metrics (accuracy, overall error, error coded without NA, NA coded & gender bias) are common in previous studies in gender detection (*Wais, 2016*; *Santamaría & Mihaljević, 2018*). Accuracy is an overall measure to evaluate a classification model. It is computed as the ratio between correct predictions and total predictions. 'Overall error' is the ratio between all the errors and the total of predictions. Non-classifications are considered as errors. Overall error is usually called 'error coded' in gender inference studies. 'Error coded without NA' measures the total number of misclassifications over the

total number of predictions without considering non-classifications. 'NA coded' is the ratio between the number of non-classifications and the total number of predictions. Usually a good gender-inference method aims to get a low value of error coded without NA while also keeping low the ratio of non-classifications. 'Error gender bias' measures the direction and the intensity of the prediction. If negative, the estimated number of males is higher than in the real data. If positive, the estimated number of females is higher. The closer to 0, the better.

For this study we also computed Cohen's kappa and Krippendorff's alpha which are commonly used as measures of inter-coder reliability. To our best knowledge this study is the first instance in which they are used to assess results of gender-inference. Cohen's kappa quantifies the level of agreement between two raters considering the probability of agreement occurring by chance. For classifiers, Cohen's kappa can be used to compare their performance against the ground truth. Krippendorff's alpha is similar to Cohen's kappa taking into consideration non-classifications. For this study, it was computed using the K-alpha calculator (*Marzi, Balzano & Marchiori, 2024*). Cohen's kappa and Krippendorff's alpha return values ranging from −1 to 1, with −1 representing systematic disagreement, 0 random guess and one unanimous agreement. Values are commonly interpreted as no-agreement (<0), slight (0.01–0.20), fair (0.20–0.40), moderate (0.41–0.60), substantial (0.61–0.80) or high (0.81–1.00) agreement (*McHugh, 2012*).

F1-score is a machine learning metric commonly preferred when the dataset is unbalanced and both types of misclassifications are equally important. Since we have two classes with no true positives or true negatives, we computed the macro F1-score, which is the mean of F1-score for each class (female and male) independently. It provides a single performance metric that summarizes how well the prediction is doing across all classes.

## Procedure

The foundation of ChatGPT is a large-scale, non-deterministic language model, which inherently introduces variability in the output, even when the input remains constant. Additionally, the model's output is influenced by various factors, including the nature of the prompt used. To evaluate the stability of ChatGPT and compare its performance with GDTs, we devised and executed a series of tests.

Firstly, we focused on stability testing by conducting 20 iterations of tests using identical input (presented in Table 1) for both versions of ChatGPT. The results were subjected to statistical analysis to calculate the standard deviation and confidence intervals of the target metrics. A low degree of variability and tight confidence intervals are indicative of stability, while the opposite could be indicative of instability even for consistent input. The outcomes of these tests also served as a comparative measure of performance between versions of ChatGPT using one-way ANOVA tests. It is important to note that GDTs, such as Namsor and Gender-API, were excluded from this initial phase. Their deterministic nature, which ensures a consistent result, precludes the possibility of collecting a diverse sample set for variability computation.

Subsequently, we assessed the performance of ChatGPT under varying prompting conditions. This approach allowed us to evaluate the model's adaptability and effectiveness

**Table 3 Prompts tested to assess the performance of ChatGPT 3.5 and 4o.**

| # | Prompt |
|---|--------|
| 0 | `{full_name}. \n\n On a first and last name basis, give me a letter answer only, f if female; m if male.\n\n` |
| | **Description: Base case selected after initial exploration.** |
| 1 | `Full name: {full_name}.\nOn a first and last name basis, give me a letter answer only, f if female; m if male.\n\n` |
| | **Description: Full name is provided first to analyze position of information** |
| 2 | `Full name: {full_name}.On a first and last name basis, give me a letter answer only, w if woman; m if man.` |
| | **Description: Ask for woman or man instead of male or female** |
| 3 | `Full name: {full_name}.On a first and last name basis, give me a letter answer only, g if girl; b if boy.` |
| | **Description: Ask for girl or boy instead of male or female** |
| 4 | `First name: {first_name}, middle name: {middle_name}, last name: {last_name}.On a first and last name basis, give me a letter answer only, f if female; m if male.` |
| | **Description: Separate first name and last name, and include middle name** |
| 5 | `First name: {first_name}.On a first name basis, give me a letter answer only, f if female; m if male.` |
| | **Description: Use only first name** |

**Note:**
For each prompt the names between brackets represent names of fields of the dataset. 'Description' includes the variation over the base case presented as Prompt0.

across diverse inputs. Prompts were selected based on their potential to test the model's response to variations in instructions, input information, and output format. Table 3 presents the prompts selected for testing in this study. Prompt 0 represents a base case that was chosen after initial exploration. Prompts 1 changes the instructions given to the model by giving clear initial indication of the name. Prompts 2 to 3 focus on the output by selecting different wording and output formats, such as boy/girl. Prompts 4 to 5 focus on the input information, presenting it in two different ways: separating first and last name, and including middle name when available (prompt 4), and using only the first name (prompt 5). Although possible options are virtually infinite, we tried to define a narrow subset to test plausible variations. If previous stability tests show that variability is low for the same input, then prompts should be tested multiple times. However, since previous tests showed a significant degree of stability (see Results section), we decided to run each prompt three times and take the median of the three runs to get more robust measures of ChatGPT performance. This approach mitigates the impact of outliers and ensures that the results are not skewed by a single anomalous run, even if standard deviation is low. Furthermore, by repeating the tests, we increase the reliability of the results, providing a more accurate representation of the behavior under different conditions. This method, although time- and resource-consuming, allows for a more comprehensive analysis and contributes to the validity of our findings.

In the final stage, we compared the results of ChatGPT with two GDTs, Namsor and Gender-API. The aim of the study is to compare the performance of gender detection in the most stringent and challenging scenario, so we opted for inference from a full non-split name, in case of similar performance with separated first and last names. Otherwise, the comparison should include both cases for all tools. For ChatGPT, we selected the best performer based on the results of the previous stability and performance tests across all evaluation metrics. For the GDTs, we considered both cases with and without calibrated

**Table 4 Confusion matrices for the stability tests of ChatGPT versions.**

| Version | | Male_predicted | | Female_predicted | | Unknown_predicted | |
|---|---|---|---|---|---|---|---|
| | True class | M | SD | M | SD | M | SD |
| ChatGPT 3.5 | Male | 3,667.1 | 6.6 | 143.7 | 6.5 | 0.3 | 0.4 |
| | Female | 190.5 | 4.2 | 1,777.3 | 4.4 | 0.3 | 0.4 |
| ChatGPT 4o | Male | 3,688.8 | 7.1 | 117.1 | 6.5 | 5.1 | 2.4 |
| | Female | 121.9 | 4.9 | 1,843.3 | 4.5 | 2.9 | 1.8 |

**Note:**
  Mean and standard deviations for 20 runs.

probability, as previously described. This final comparison included six different cases, which correspond to the two GDTs with and without calibrated probability (Namsor, $Namsor_{p > 0.55}$, GenderAPI, and Gender-API$_{acc > 55}$), and the best performers of the two versions of ChatGPT (ChatGPT 4o and ChatGPT 3.5). These cases were chosen to provide a comprehensive comparison across different tools and settings. The final dataset including the results of the stability runs, prompt variations, and GDTs' outcomes are included as Supplemental Material.

# RESULTS

## ChatGPT stability tests

To test the stability of the models, we ran the base case twenty times. The mean and standard deviation of the six values in the confusion matrix are presented in Table 4. Results show that ChatGPT 4o predicts both males and females more accurately than its predecessor. Specifically, ChatGPT 4o demonstrates improved consistency in predicting females compared to ChatGPT 3.5. This is reflected in the higher average number of females correctly classified (1,843.3 compared to 1,777.3) and the lower average number of females incorrectly classified as males (121.9 compared to 190.5). While ChatGPT 3.5 classifies almost all instances, ChatGPT 4o returns a small number of non-classifications (averaging 5.1 for males and 2.9 for females).

Table 5 shows the results the metrics for the stability tests calculated over twenty runs of the base case. While both versions performed well, ChatGPT 4o showed a slight improvement in all metrics resulting in more accuracy, F1-score, Cohen's kappa and Krippendorff's alpha (Fig. 3) while also presenting less errors and gender bias. For instance, the accuracy of ChatGPT 4o was 0.957, slightly higher than the 0.942 of ChatGPT 3.5. Similarly, the F1-score for ChatGPT 4o was 0.954, compared to 0.935 for ChatGPT 3.5. However, it's important to note that these differences, while statistically significant, are quite small, around 1.5% for accuracy, 1.7% for F1-score, and 3.8% for Cohen's kappa and Krippendorff's alpha. Since this last two metrics are more stringent, the differences in favor of ChatGPT 4o suggest that its performance is more consistent and reliable. In terms of error coding (Fig. 4), ChatGPT 4o also performed slightly better, with an average error of 0.043 compared to 0.058 for ChatGPT 3.5. When excluding non-classifications (NAs) from the error calculation, the averages remain similar (0.041 for ChatGPT 4o and 0.058 for ChatGPT 3.5). Interestingly, ChatGPT 4o returned a small number of non-

**Table 5 Metrics of stability tests with ChatGPT versions.**

| Metric | Version | M | SD | Mdn | 95% CI | F |
|---|---|---|---|---|---|---|
| Accuracy | ChatGPT 3.5 | 0.942 | 0.001 | 0.942 | [0.941–0.943] | 1,199** |
| | ChatGPT 4o | 0.957 | 0.002 | 0.957 | [0.957–0.958] | |
| Error coded | ChatGPT 3.5 | 0.058 | 0.001 | 0.058 | [0.057–0.059] | 1,199** |
| | ChatGPT 4o | 0.043 | 0.002 | 0.043 | [0.042–0.043] | |
| Error coded (without NA) | ChatGPT 3.5 | 0.058 | 0.001 | 0.058 | [0.057–0.058] | 1,431** |
| | ChatGPT 4o | 0.041 | 0.002 | 0.041 | [0.041–0.042] | |
| NA coded | ChatGPT 3.5 | 0.000 | 0.000 | 0.000 | [0.000–0.000] | 146** |
| | ChatGPT 4o | 0.001 | 0.000 | 0.001 | [0.001–0.002] | |
| Gender bias | ChatGPT 3.5 | −0.008 | 0.001 | −0.009 | [−0.008 to −0.007] | 275** |
| | ChatGPT 4o | −0.001 | 0.001 | −0.001 | [0.002–0.000] | |
| Cohen's kappa | ChatGPT 3.5 | 0.870 | 0.003 | 0.870 | [0.870–0.872] | 1,522** |
| | ChatGPT 4o | 0.908 | 0.003 | 0.909 | [0.906–0.909] | |
| Krippendorff's alpha | ChatGPT 3.5 | 0.870 | 0.003 | 0.870 | [0.869–0.872] | 1,523** |
| | ChatGPT 4o | 0.908 | 0.003 | 0.909 | [0.906–0.909] | |
| F1-score | ChatGPT 3.5 | 0.935 | 0.001 | 0.933 | [0.935–0.936] | 1,523** |
| | ChatGPT 4o | 0.954 | 0.002 | 0.954 | [0.953, 0.955] | |

**Notes:**
F value of one-way ANOVA tests F(1,38).
**$p < 0.01$.

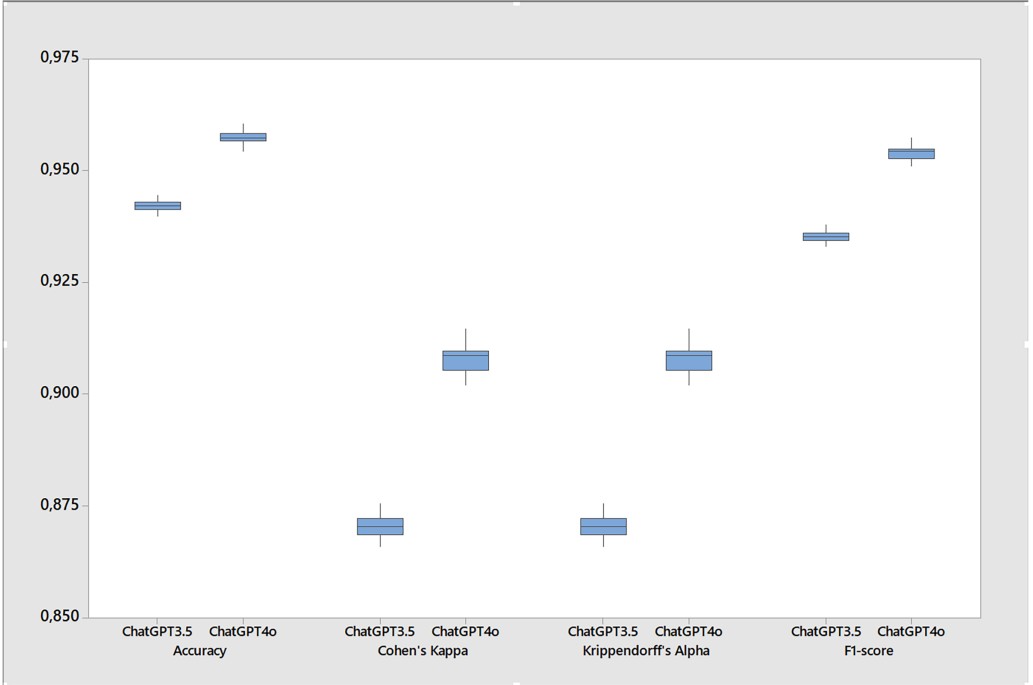

**Figure 3 Boxplot of performance metrics for the stability test comparing ChatGPT 3.5 and ChatGPT 4o (20 runs in each version).**

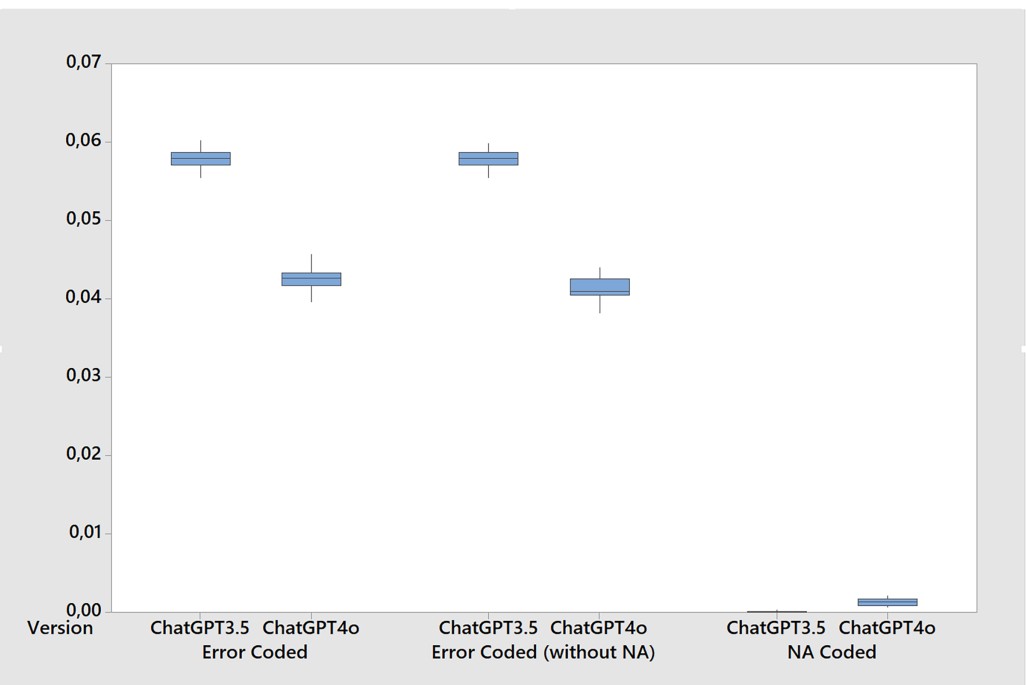

**Figure 4** Boxplot of error metrics for the stability tests comparing ChatGPT 3.5 and ChatGPT 4o (20 runs in each version).

classifications (0.001), while ChatGPT 3.5 returned virtually none (0.000) across all test cases. In terms of gender bias, both versions showed a slight bias towards misclassifying females as males, but again, the difference was small ($-0.001$ for ChatGPT 4o and $-0.008$ for ChatGPT 3.5). Finally, both versions showed high agreement with the true classifications, as indicated by the Cohen's kappa and Krippendorff's alpha metrics. However, ChatGPT 4o had slightly higher agreement (0.908 for both metrics) compared to ChatGPT 3.5 (0.870 for both metrics). So, while ChatGPT 4o shows a slight improvement in performance over ChatGPT 3.5, the differences are small although statistically significant across all metrics. Further, small standard deviations and tight confidence intervals across all metrics in both versions suggests that ChatGPT is stable for inferring gender.

## ChatGPT prompt tests

The performance metrics of the two versions of ChatGPT across six different prompts provide further insights into their performance characteristics (Table 6). Firstly, it is observed that ChatGPT 3.5 exhibits superior performance when the name is clearly indicated at the beginning of the query (Prompt1). This is evidenced by the highest accuracy of 0.960, the lowest error coded of 0.040, and the highest Cohen's kappa and Krippendorff's alpha of 0.911. This suggests that the model's ability to understand and respond accurately is enhanced when the context is explicitly defined at the beginning of the prompt. Secondly, ChatGPT 4o demonstrates a remarkable stability across most performance metrics. The accuracy, error coded, Cohen's kappa, Krippendorff's alpha, and

**Table 6 Performance metrics for six different prompts of ChatGPT versions.**

| | ChatGPT 3.5 | | | | | |
|---|---|---|---|---|---|---|
| Metric | Prompt0 | Prompt1 | Prompt2 | Prompt3 | Prompt4 | Prompt5 |
| Accuracy | 0.942 | **0.960** | 0.955 | 0.957 | 0.949 | 0.939 |
| Error coded | 0.058 | **0.040** | 0.045 | 0.043 | 0.051 | 0.061 |
| Error coded without NA | 0.058 | **0.040** | 0.045 | 0.043 | 0.051 | 0.061 |
| NA coded | 0.000 | 0.000 | 0.000 | 0.000 | 0.000 | 0.000 |
| Gender bias | −0.008 | −0.012 | −0.024 | −0.010 | −0.009 | **−0.003** |
| Cohen's kappa | 0.870 | **0.911** | 0.898 | 0.905 | 0.887 | 0.864 |
| Krippendorff's alpha | 0.870 | **0.911** | 0.898 | 0.905 | 0.887 | 0.864 |
| F1-score | 0.935 | **0.955** | 0.949 | 0.952 | 0.943 | 0.932 |
| | ChatGPT 4o | | | | | |
| Metric | Prompt0 | Prompt1 | Prompt2 | Prompt3 | Prompt4 | Prompt5 |
| Accuracy | 0.957 | 0.957 | **0.961** | 0.961 | 0.955 | 0.928 |
| Error coded | 0.043 | 0.043 | **0.039** | 0.039 | 0.045 | 0.072 |
| Error coded without NA | 0.041 | 0.042 | **0.039** | 0.039 | 0.045 | 0.07 |
| NA coded | 0.001 | 0.001 | 0.000 | 0.000 | 0.000 | 0.002 |
| Gender bias | **−0.001** | 0.003 | −0.007 | −0.008 | −0.001 | 0.024 |
| Cohen's kappa | 0.908 | 0.906 | **0.913** | 0.912 | 0.900 | 0.846 |
| Krippendorff's alpha | 0.908 | 0.906 | **0.913** | 0.912 | 0.900 | 0.846 |
| F1-score | 0.954 | 0.953 | **0.957** | 0.956 | 0.950 | 0.923 |

Note:
Prompt0 presents the average of 20 runs. Other prompts were run once. Differences between Cohen's kappa and Krippendorff's alpha are less than 0.001 in all cases. The best result for each metric is highlighted in bold.

F1-score are consistently high across all prompts, indicating the model's robustness and reliability in various contexts. The slight variations in these metrics across different prompts are within an acceptable range of variability, further emphasizing the model's stability. Thirdly, it is noteworthy that ChatGPT3.5 performs better than ChatGPT 4o when only the first name is given (Prompt5) and results are comparable with other prompts. For instance, ChatGPT 3.5 achieves an accuracy of 0.939 and an F1-score of 0.932, while ChatGPT 4o reaches an accuracy of 0.928 and an F1-score of 0.923. This indicates that the models are capable of generating accurate and relevant responses even with minimal context, demonstrating their flexibility and adaptability.

The gender bias metric is negative for most prompts in both versions of ChatGPT, indicating a slight bias towards incorrectly classifying females as males. However, it is interesting to note that for ChatGPT 4o with Prompt5 (using only first name), the gender bias is positive (0.024), which is a deviation from the trend observed in other prompts. This suggests that the way the prompt is structured could potentially influence the gender bias in the model's responses. The NA Coded metric is zero for all prompts in ChatGPT 3.5, indicating that there were no instances where the model failed to generate a response. However, in ChatGPT 4o, there were a few instances (Prompt0 and Prompt1) where the model did not generate a response. So, the number of non-classifications is also remarkably

low. Both versions of ChatGPT show a drop in performance metrics (Accuracy, Cohen's kappa, Krippendorff's alpha, and F1-score) for Prompt5 compared to other prompts. This could suggest that the models have difficulty in generating accurate and relevant responses when only the first name is given. For both versions of ChatGPT, the Cohen's kappa and Krippendorff's alpha values in Table 6 are identical for each prompt. Although there are differences, in all cases are less than 0.001 and not reflected in the table. This consistency indicates that both these metrics, which measure the agreement between raters, are giving us a reliable measure of the model's performance.

In conclusion, these findings highlight the strengths and capabilities of ChatGPT 3.5 and 4o, providing valuable insights for recommended improvements and adaptations. For ChatGPT 3.5 the models' performance can be optimized by clearly indicating the name at the beginning of the query. When providing only the first name, ChatGPT 3.5 performs better than ChatGPT 4o and results are similar to other prompts. Furthermore, the stability of ChatGPT 4o across most performance metrics underscores its reliability and robustness in various contexts.

## ChatGPT *vs* gender APIs

In this final subsection of the results, we compare the performance of ChatGPT with GDTs. For both versions of ChatGPT, we select the best performers from the previous tests: For ChatGPT3.5 the prompt in which the full name is provided at the beginning (prompt1 in Table 3), and for ChatGPT4o the prompt that asks whether the full name is of a woman or man instead of male and female (prompt2 in Table 3). For the GDTs, we include Namsor and Gender-API with and without calibrated probability.

Confusion matrices are presented in Table 7. We can see that in all cases most instances are correctly classified. Also, Namsor and ChatGPT 3.5 can predict all instances while ChatGPT 4o returns only 1 non-classification. Performance metrics (Table 8) show that in terms of overall accuracy both versions of ChatGPT outperform all other inference methods classifying correctly 96% of all instances, although difference is around a 3% when compared to GDT: Namsor (93%) and Gender-API (92.8%). For Cohen's kappa ChatGPT slightly overperforms GDTs with both versions of returning over 0.91 while Namsor and Gender-API obtain 0.846 and 0.841 respectively, although high values show a a high level of reliability in all the cases. When GDTs are calibrated by probability, their overall performance decreases as the number of non-classification increases. We can observe that the number of non-classifications (NA coded) is 5.6% for $Namsor_{p > 0.55}$ and 2.6% for $Gender-API_{acc > 55}$. However, when we compare performance in this scenario, as measured by 'Error coded without NA', we can see that results are comparable. ChatGPT leads with around 4% in both versions, followed by $Namsor_{p > 0.55}$ with 4.8% and $Gender-API_{acc > 55}$ with 5.4%. Krippendorff's alpha further confirms these results with both versions of ChatGPT getting over 0.910, followed by $Namsor_{p > 0.55}$ with 0.895 and $Gender-API_{acc > 55}$ with 0.873. So, metrics of inter-coder agreement show that there are no important differences in terms of reliability when comparing ChatGPT, Namsor and Gender-API when the GDTs' outcomes are calibrated by the probability of the prediction. Values higher than 0.8 suggest a high level of agreement. However, this comes at the

**Table 7 Confusion matrices for Namsor, Gender-API and ChatGPT.**

| Inference tool | True class | Male$_{predicted}$ | Female$_{predicted}$ | Unknown$_{predicted}$ |
|---|---|---|---|---|
| Namsor | Male | 3,553 | 258 | 0 |
| | Female | 148 | 1,820 | 0 |
| Namsor | Male | 3,447 | 149 | 215 |
| (Prob. > 0.55) | Female | 111 | 1,757 | 100 |
| Gender-API | Male | 3,619 | 153 | 39 |
| | Female | 206 | 1,745 | 17 |
| Gender-API | Male | 3,588 | 126 | 97 |
| (Accuracy > 55) | Female | 192 | 1,720 | 56 |
| ChatGPT 3.5 | Male | 3,732 | 79 | 0 |
| (Prompt1) | Female | 150 | 1,818 | 0 |
| ChatGPT 4o | Male | 3,718 | 93 | 0 |
| (Prompt2) | Female | 131 | 1,836 | 1 |

**Note:**
'Namsor' and 'Gender-API' present results without using the calibrated probability or accuracy as returned by the service. 'Namsor$_{p > 0.55}$' presents metrics when probability is calibrated to more than 0.55. 'Gender-API$_{acc > 55}$' presents metrics calibrated to over 55 as returned by the service. 'ChatGPT 3.5' and 'ChatGPT 4o' present results for the best performing prompt of previous tests.

**Table 8 Performance metrics for Namsor, Gender-API and ChatGPT.**

| Metric | Namsor | Namsor$_{p > 0.55}$ | Gender-API | Gender-API$_{acc > 55}$ | ChatGPT 3.5 | ChatGPT 4o |
|---|---|---|---|---|---|---|
| Accuracy | 0.930 | 0.901 | 0.928 | 0.919 | 0.960 | 0.961 |
| Error coded | 0.070 | 0.099 | 0.072 | 0.081 | 0.040 | 0.039 |
| Error coded without NA | 0.070 | 0.048 | 0.063 | 0.054 | 0.040 | 0.039 |
| NA coded | 0.000 | 0.056 | 0.010 | 0.026 | 0.000 | 0.000 |
| Gender bias | 0.02 | 0.007 | −0.009 | −0.012 | −0.012 | −0.007 |
| Cohen's kappa | 0.846 | 0.893 | 0.841 | 0.823 | 0.911 | 0.913 |
| Krippendorff's alpha | 0.846 | 0.895 | 0.859 | 0.873 | 0.911 | 0.913 |
| F1 score | 0.923 | 0.947 | 0.930 | 0.936 | 0.955 | 0.957 |

**Note:**
Columns Namsor and Gender-API present the results of metrics using gender inferred without calibration by probability or accuracy. Column 'Namsor$_{p > 0.55}$' presents metrics when probability is calibrated to more than 0.55. Column 'Gender-API$_{acc > 55}$' presents metrics calibrated to over 55 as returned by the service. 'ChatGPT 3.5' and 'ChatGPT 4o' present results for the best performing prompt of previous tests.

expense of a higher number of non-classifications for GDTs, which lies between 2.6% and 5.6%, as compared to ChatGPT which codes all instances.

# DISCUSSION AND CONCLUSION

Our study advances the literature on gender inferences by testing the stability and reliability of ChatGPT for gender detection and by comparing the performance of two GDTs and ChatGPT against a dataset of ground truth data, providing intercoder reliability statistics to test their predictive performance. Different from previous research on ChatGPT for gender detection (*Alexopoulos et al., 2023*; *Sebo, 2024*), our study empirically demonstrates the stability of two versions of the popular chatbot for gender detection. Further, we also analyze the performance of ChatGPT under different prompting

conditions that test the wording, input and output of the inference. Also new from previous research on automatic gender inference (*Bérubé et al., 2020*; *Sebo, 2021a*, *2021b*, *2022*; *VanHelene et al., 2024*), our study implements a more stringent comparison (including inter-coder reliability statistics with Kohen's kappa and Krippendorff's alpha) to offer more robust empirical evidence and research suggestions that may resonate to future scholars interested in gender inference from names. Although *Alexopoulos et al. (2023)* compared ChatGPT to GDTs on a significantly larger dataset; they reported exclusively machine learning metrics which, in our opinion, are not appropriate for gender inference since for this classification problem there are no false positives or false negatives. Further, their study did not consider non-classifications and calibrated probabilities. Our findings support and extend previous study by *Sebo (2024)*. Our study also presents methodological verifiable approach to compare all tools under similar conditions with a dataset that was previously used in existing literature focusing on the most challenging version of the gender inference problem, which is the genderization from a full name.

In a series of 20 runs using a large and diverse dataset, we demonstrated that ChatGPT exhibits high stability, as evidenced by a very low standard deviation and tight confidence intervals across key performance metrics and error rates. This stability is particularly noteworthy given the sample size of our tests (20 runs). Moreover, in most runs, the number of instances where ChatGPT failed to classify the input (non-classifications) was zero. We also found that the wording, input, and output variations in the prompts had a significant impact on performance, although this impact was limited to a 2.1% difference in accuracy between the best and worst performers in ChatGPT 3.5 and a 3.3% difference in ChatGPT 4o. Interestingly, when we excluded prompts where only the first name was provided, the difference in accuracy for ChatGPT 4o reduced to just 0.6% among all comparable prompts. When comparing ChatGPT to GDTs, our findings suggest that the overall accuracy of ChatGPT 3.5 and ChatGPT 4o is slightly better than the other two gender detection tools, although this improvement is only around a 3%, which suggest that the performance of the two GDTs and ChatGPT artificial intelligence model are fairly similar when it comes to gender predictions. However, it must also be noted that ChatGPT also offers a very low number of non-classifications compared to the other two GDTs, thus turning this generative language model into a solid alternative for gender inference. Both versions of ChatGPT (3.5 and 4o) yield comparable results. Therefore, researchers who aim to classify gender from names using ChatGPT could consider either version. Although future iterations may exhibit progressive improvements, the potential for achieving perfect accuracy is only around 4%. Moreover, ChatGPT 3.5 might be a more cost-effective choice due to its lower operational cost, which is 10 times less than the cost of operating ChatGPT 4o.

Our tests also show that when results are calibrated by probability, the outcome is slightly different. Although ChatGPT still offers the best results as measured, arguably, by the most stringent statistical test, *i.e.*, the Krippendorff alpha (>0.91), Namsor is very close (0.895) and ChatGPT is also comparable (0.873). This, however, comes at the expense of a higher number of non-classifications for GDTs (5.6% for Namsor and 2.6% for Gender-

API) which impacts on the overall accuracy. We may conclude that the reliability of the two GDTs and ChatGPT 4o is rather similar and comparable in this case.

In this regard, our study emphasizes the need to report probability or accuracy predictions as they stand as robust methical mechanisms through which journal editors, reviewers, and readers can evaluate the output of automatic gender detection tools. Accordingly, gender detection tools such as Namsor or Gender-API may be preferred alternatives in those cases in which such statistical analyses maybe needed or mandatory, like research studies. For more general commercial applications of gender classifications, the latest version of ChatGPT may meet the necessary requirements. In short, our study concludes that despite the reliable gender inferences offered by ChatGPT, the accuracy statistics transparently provided by Namsor warrant its use for empirically based research enterprises in which research may need to control and fine tune the output of this gender detection tool. Although ChatGPT 4o does not offer any measure of accuracy to calibrate the predictions, its high accuracy in combination with the low number of non-classifications, which is zero in most tests, demonstrates that it presents a solid alternative for gender inference. Well-established GDTs like Namsor and Gender-API get close to ChatGPT's performance only when probability is calibrated. However, and justifiably, the possibility of calibrating results poses a strong methodological reason for preferring GDTs over ChatGPT. For example, for scientific studies in which it is necessary to control the error of gender inference, researchers should opt for GDTs. For general cases and commercial applications, users should prefer ChatGPT since it infers most cases with a high degree of accuracy.

The study presents several limitations. It uses tools that categorize gender as either male or female, failing to capture the non-binary nature of gender. This could potentially marginalize nonbinary or transgender individuals and limit the applicability of these tools. The study also relies on manually labeled gender, which introduces the potential for human bias or error into the dataset. The accuracy of the tools might be affected by the quality of the manual labeling. The study does not specifically address the performance of the tools on Asian names. Given the diversity and complexity of Asian names, the tools might exhibit lower accuracy for these names as previous work already pointed (*Sebo, 2022*). The study was conducted using a specific dataset, which may not be representative of other populations. Future studies might also investigate the research question of how the origin of names affects the performance of gender detection tools. While our current study acknowledges the limitations and potential biases associated with inferred nationality data, further research could explore this aspect more deeply. Utilizing dedicated tools like Namsor Onomastics to determine the origin of names could provide valuable insights. Such investigations could help to better understand and mitigate the biases and uncertainties in gender detection, particularly for names from diverse cultural backgrounds, including Asian names. This would contribute to the development of more accurate and inclusive gender detection methodologies.

Furthermore, the study's findings are based on a specific version of ChatGPT and two specific GDTs. The results might not be generalizable to other versions of ChatGPT or other GDTs. While the study found that ChatGPT exhibits high stability, it's based on a

limited number of runs. More extensive testing might be needed to confirm this finding. The study found that the wording, input, and output variations in the prompts had an impact on performance. This suggests that the results might vary depending on how the prompts are structured. Further, ChatGPT offers a very low number of non-classifications compared to the other two GDTs. However, this could also be seen as a limitation as it might lead to overconfidence in the results. These limitations highlight the complexity of gender inference and the challenges in developing tools that can accurately and ethically perform this task. It's important for future research to address these limitations and strive for more inclusive and accurate gender detection methods.

In conclusion, while the study demonstrates the potential of ChatGPT for gender inference, it also highlights several limitations that need to be considered. The performance of ChatGPT and GDTs like Namsor and Gender-API are fairly similar, with slight variations depending on the conditions and calibration of results. However, the ease of use and accessibility of GDTs might make them a preferred choice for many users. Specifically, Namsor and Gender-API allow users to upload a CSV file of names and enrich it with an additional "gender" and calibrated probability columns, a feature that ChatGPT does not currently offer. This simplicity and user-friendliness, combined with the ability to control and fine-tune the output, make GDTs a solid choice for gender inference, especially for users without extensive computer knowledge. In terms of pricing, ChatGPT offers a cost-effective alternative for large datasets for technologically-oriented users. Meanwhile, GDTs, particularly NamSor, provide a free user-friendly alternative for smaller datasets. As the field of AI continues to evolve, it will be interesting to see how these tools develop and how their performance and features compare in the future. It's crucial that future research continues to address the identified limitations and strives for more inclusive and accurate gender detection methods.

### Funding
The authors received no funding for this work.

### Competing Interests
The authors declare that they have no competing interests.

### Author Contributions
- Adrián Domínguez-Díaz conceived and designed the experiments, performed the experiments, analyzed the data, performed the computation work, prepared figures and/or tables, authored or reviewed drafts of the article, and approved the final draft.
- Manuel Goyanes conceived and designed the experiments, performed the experiments, analyzed the data, authored or reviewed drafts of the article, and approved the final draft.
- Luis de-Marcos conceived and designed the experiments, performed the experiments, analyzed the data, prepared figures and/or tables, authored or reviewed drafts of the article, and approved the final draft.

- Víctor Pablo Prado-Sánchez conceived and designed the experiments, performed the computation work, prepared figures and/or tables, authored or reviewed drafts of the article, and approved the final draft.

## Data Availability

The raw data and code are available in the Supplemental Files.

## Supplemental Information

Supplemental information for this article can be found online at http://dx.doi.org/10.7717/peerj-cs.2378#supplemental-information.

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
