# Peer review of "Comparative analysis of automatic gender detection from names: evaluating the stability and performance of ChatGPT versus Namsor, and Gender-API"

_PeerJ Computer Science, doi:10.7717/peerj-cs.2378_

## Round 0.1 · original submission · Major Revisions

Please respond in detail to the reviewer comments

Reviewer 1 ·

Basic reporting

see below

Experimental design

see below

Validity of the findings

see below

Additional comments

It was with great interest that I read this interesting study by Dominguez and colleagues, which compares the performance of several gender detection tools and ChatGPT version 3.5 and 4.0 in determining gender from individuals' names.

The main reservations I would have with this study are as follows:

A. The results are not really new, several studies having already demonstrated that Gender API, Namsor, chatGPT 3.5 and ChatGPT 4.0 were accurate for determining gender from individuals' names. The authors have included references for the studies evaluating Gender API and Namsor, but not for the study evaluating ChatGPT 3.5 and ChatGPT 4.0 (see below for the reference).

B. The authors compared the tools but did not perform statistical tests to assess whether the differences observed were statistically significant (confidence intervals and p-values are missing).

C. The authors did not describe in detail the database used for the study. In particular, the reader would be interested to know the origin of the individuals. Several studies have shown that gender detection tools perform poorly with certain names, such as Asian names. It would probably be interesting to use a tool estimating the probable origin of individuals, as has been done in other previous studies. Several such tools have been validated for this purpose, e.g. Namsor Onomastics.

D. Finally, the reader does not know how the actual gender of the individuals was estimated. Was the actual gender of the individuals “self-determined”?

E. For this study to add value to what is already known, the authors could, for example, estimate the stability of the results for ChatGPT 3.5 and ChatGPT 4.0 (see below). The authors could also evaluate the performance of the tools depending on whether nationality or estimated origin of individuals is added, or whether only first name vs. first name and surname is tested. Finally, the authors could evaluate the performance of ChatGPT with various prompts. Does the way the question is asked affect the tool's performance?

I have several more specific comments for the authors:

1) The authors should provide the complete study database with any additional analyses.

2) abstract: please explain what is meant by “unsplit full name”.

3) abstract: the authors compare the results of different tools using different performance metrics, but lack statistical tests with confidence intervals and p-values. Are the differences observed statistically significant?

4) abstract “yet GDTs (especially Namsor) should be used for research-oriented purposes”. If ChatGPT is at least as powerful as Namsor and Gender API, why not use ChatGPT for research purposes too?

5) Introduction: it would be useful to briefly describe a few studies that have used Namsor, Gender API and ChatGPT (study objective, main results).

6) Introduction “although a few studies report and compare against the truth, they mostly focus on comparing general GDT's performance [...]. That is also the aim of this study, isn't it? In other words, what does this study add to what is already known?

7) Introduction “Secondly, [...] unexplored the potential power of [...] ChatGPT in predicting gender from names”. This is not entirely true, as a recent study showed that the performance of ChatGPT 3.5 and 4.0 was excellent (ref: Sebo P. What Is the Performance of ChatGPT in Determining the Gender of Individuals Based on Their First and Last Names? JMIR AI 2024;3:e53656, doi: 10.2196/53656). What does this study add to what was shown in the study conducted by Sebo ?

8) Introduction: the authors have added two performance tests (Krippendorff alpha and Cohen's kappa). What do these two tests add over and above those usually used to assess the performance of gender detection tools?

9) Introduction: in addition, should not “alpha” and “kappa” be in lower case?

10) Methods: authors should describe how the actual gender of individuals in their database was determined. Was the gender determined by “self-identification”? If not, this is an important limitation of the study.

11) Methods: if the actual gender of the individuals in the database was determined by “self-identification”, how many individuals in the database did not consider themselves either female or male?

12) Methods: do the authors have data on the origin of names in their database? If not, have they considered determining the origin of names using a dedicated tool, such as Namsor Onomastics ?This tool has recently been validated for this purpose.

13) Methods: have the authors examined whether the performance of the tools depends on the origin of the names? Several studies have shown, for example, that gender detection tools perform less well with Asian names.

14) Methods: the authors should explain why they chose Namsor and Gender API as their gender detection tools.

15) Methods “they are the two most used name-to-gender online services”. The authors should add a reference to support this statement.

16) Methods: the structure of the second part of the methods section could be improved. For example, the authors could describe each of the tools in turn. The authors could also add two screenshots for each tool, showing the reader (a) how to use the tool in practice and (2) how the results are given by the tool.

17) Methods “the initial four metrics”. The authors should list these four metrics

18) Methods “formulas are provided in Table 1”. Authors should provide formulas for all metrics used in the study.

19) Methods “ [...] no-agreement (<0), slight (0.01-0.20), fair (0.20-0.40) [...]”. Authors should add a reference for these definitions.

20) Results: authors should perform statistical tests and provide confidence intervals and p-values, as the differences observed in the study are not necessarily statistically significant.

21) Results “ChatGPT outperforms all other [...]. The authors should specify that this is ChatGPT 4.0.

22) Results “we can also observe that ChatGPT 3.5 performs poorly”. The authors should repeat the analyses at least once or twice for ChatGPT 4.0 and ChatGPT 3.5 to verify the stability of the results (ref: Zhu L, Mou W, Yang T, Chen R. ChatGPT can pass the AHA exams: Open-ended questions outperform multiple-choice format. Resuscitation. 2023 Jul;188:109783. doi: 10.1016/j.resuscitation.2023.109783. PMID: 37349064).

23) Results: these results contradict Sebo's study, which showed that both ChatGPT 3.5 and 4.0 were accurate (ref: Sebo P. What Is the Performance of ChatGPT in Determining the Gender of Individuals Based on Their First and Last Names? JMIR AI 2024;3:e53656, doi: 10.2196/53656). If the results obtained by Dominguez and colleagues in their study are stable for ChatGPT 3.5 and 4.0, the authors should try to explain why their results are different from those of Sebo.

24) Results “although ChatGPT 4.0 does not offer [...] high degree of accuracy”. This paragraph should be moved to the discussion section.

25) Discussion: a paragraph on the study's limitations is missing.

26) Discussion: a concluding paragraph is missing. Gender detection tools can be preferred to ChatGPT as they are easy to use and do not require extensive computer knowledge. With Namsor and Gender API, you can easily upload a csv file of names. The file is then enriched with an additional “gender” column. ChatGPT does not (yet) offer this option.

Reviewer 2 ·

Basic reporting

This paper has clear results that are well reported. However, the introduction section and discussion and conclusions section have several overly flowering terms and expressions that lead me to wonder if those sections may have been edited by a generative AI tool. Example phrases that are overly flowering include: “Extant research has been highly prolific in testing….” (line 40); “potential limitations in their inference muscle” (line 45); “research gaps remain unchartered” (line 46); “or are circumscribed to specific geographies” (line 52); “Stricto senso, our findings first suggest ..." (line 212); “although the amelioration is only less that a 2%” (line 214 – note, also, it should be “less than 2%”).

There are some lines that are unclear (e.g. line 50: “the literature may introduce bias in their recommendations”).

It is important to acknowledge limitations in tools that infer gender from names. These tools categorize gender as male or female (on a binary scale); however, gender exists on a non-binary scale and that is not captured by these tools.

The literature referenced is relevant; however, a paper that uses ChatGPT for gender inference is discussed in the conclusion section (Michelle, et al. 2023) but not mentioned in the introduction. In fact, in the introduction (line 59), it says, “thus remaining unexplored the potential power of generative language models, such as ChatGPT, in predicting gender from names.” The research reported in that paper (cited in the conclusions) compares ChatGPT to 3 gender inference tools on a much larger dataset, compares performance using different inputs to the tools, and by geographic region. The contribution of the research under review should be clearly articulated against the research reported on in that paper (beyond the differing measures of performance of the tools).

The raw data is supplied which is great.

There are some typos: e.g. Supplementaty should be Supplementary; “Kohen” instead of “Cohen”. Some statements should be cited: e.g., “They are the two most used name-to-gender online services.” And line 160-161 that states the common interpretations of agreement for Cohen’s Kappa and Krippendorff’s Alpha.

Experimental design

The goals of the research are clear but not overly ambitious. The contribution of the experiment and results should be more clearly articulated with respect to recent research on ChatGPT for inferring gender from names (e.g., Michelle, et al. 2023).

While the paper is clear and results well reported, the experiment itself is not very broad nor comprehensive. The dataset is not large. There are nearly twice as many names identified as male than female and no discussion of how this may impact results. There is no discussion of cost, complexity, resource use comparison of the different methods for gender identification. The contribution of using Krippendorff Alpha (called “the most stringent statistical test”) and Cohen’s Kappa is stated as unique and important. There is an opportunity to expand on this further in the discussion. There is also an opportunity to discuss issues of prompting. For example, what prompts were tried in order to get probability information from ChatGPT? What was the final prompt used? Was it the same prompt for 3.5 and 4? Were responses consistent?

Is it possible the true class rows for each tool in Table 2 are reversed? The true class males add to1968 and the true class females add to 3811 but earlier it is stated the dataset is opposite of that.

Validity of the findings

There is an opportunity to enhance the discussion of impact and novelty.

Additional comments

The experiment itself is less comprehensive than the paper cited from 2023. One option is to clearly state the novetly beyond that paper but it is not clear what that would be. Another option would be to use the unique measurements in this paper on the larger dataset available from that 2023 paper.

---

## Round 0.2 · Minor Revisions

Please follow the final review comments

Reviewer 1 ·

Basic reporting

I think the authors have greatly improved their article. However, I still have a few comments to make.

The authors reply to comment 1.D that the gender of individuals was self-determined. Yet, in response to my comment 1.10, the authors explain how their database was constructed. They state that the gender was primarily determined through internet searches using bibliographic databases, photographs, inspecting individuals' names, etc., and that the gender was manually coded by the database authors. Please check and correct this in the text if necessary.

The authors added the following sentence to the manuscript: "In a subsequent study, Sebo (2024) evaluated the performance of ChatGPT 3.5 and ChatGPT 4 using a sample of 500 names extracted from a larger database." However, upon re-reading Sebo's study, it turns out that the database contained 6,131 individuals and not 500 as mentioned by the authors. Please correct this information.

I think the authors' response to comments 1.12 and 1.13 is appropriate, but the authors should add a paragraph in the discussion mentioning that future studies might investigate this research question.

Experimental design

N/A

Validity of the findings

N/A

Additional comments

N/A

Reviewer 2 ·

Basic reporting

The additional framing of the paper to consider stability of chatGPT has greatly strengthened the paper. The authors have responded very well to issues identified in the first review.

Experimental design

The design of the new experiments are described in sufficient detail and the improved description of experiments from the earlier version of the paper are described in sufficient detail.

Validity of the findings

The findings are clear and the new tables improve the presentation of the results. The new framing and results and the description of the contributions in relation to past work show the novelty and impact of the research.

Additional comments

The authors did a great job of addressing concerns and updating the paper in light of those concerns.

There were a couple of common typos I noticed: "grounded truth" should be "ground truth" and "unknow" should be "unknown"

---

## Round 0.3 · accepted · Accept

Thank you for integrating the comments